# Gene and Protein Expression of Placental Nutrient-Stress Sensor Proteins in Fetal Growth Restriction

Elizabeth Morgan [1,2,†], Grace Chung [1,†], Seokwon Jo [1], Briana Clifton [1], Sarah A. Wernimont [2] and Emilyn U. Alejandro [1,*]

1    Department of Integrative Biology and Physiology, University of Minnesota Medical School, University of Minnesota, Minneapolis, MN 55455, USA; elmorgan@uchc.edu (E.M.); chung491@umn.edu (G.C.); joxxx057@umn.edu (S.J.); clift083@umn.edu (B.C.)

2    Department of Obstetrics, Gynecology and Women's Health, University of Minnesota Medical School, University of Minnesota, Minneapolis, MN 55455, USA; swernimo@umn.edu

\*    Correspondence: ealejand@umn.edu

†    These authors contributed equally to this work.

**Abstract:** Fetal growth restriction (FGR) and low birth weight increase the risk of non-communicable diseases such as type 2 diabetes and heart failure in adulthood. Placental insufficiency is widely considered a major contributor to FGR. Two crucial placental proteins involved in nutrient and stress sensing—O-linked N-acetylglucosamine transferase (OGT) and mechanistic target of rapamycin (mTOR) kinase—play roles in post-translational protein modification and protein translation, influencing cellular growth and metabolism in response to maternal stress, hypoxia, and nutritional status in the placenta. In our study, we examined the gene and protein profiles of OGT and mTOR in FGR and control placentae, comparing those appropriate for gestational age (AGA), while also considering potential confounding effects of fetal sex and delivery mode. Our findings revealed no significant differences in gene expression, protein levels, or activity of OGT, OGA, mTOR, or their associated markers between female AGA and FGR cesarean placentae, nor between female AGA and male AGA cesarean placentae. Additionally, the mode of delivery in female AGA placentae did not affect the levels or activity of these proteins. Overall, our study did not observe significant differences in nutrient sensor protein expression after stratifying by FGR, sex, and delivery mode. Nevertheless, these unbiased results provide a more comprehensive understanding of the complexities of placental gene expression involving OGT and mTOR.

**Keywords:** fetal growth restriction; maternal stress; placenta; nutrient sensor protein; OGT; mTOR; 11β-HSD2

## 1. Introduction

Fetal growth restriction (FGR) is defined as an overall estimated fetal weight or abdominal circumference of <10th percentile [1]. Fetal growth is dependent on a number of factors including genetics, maternal and environmental factors, placental function, and regulation of the interface between the maternal and fetal environment. One of the leading contributors in the development of FGR is alteration in placental nutrient and oxygen delivery to the fetus. Glucose and amino acids are important substrates for fetal growth, and the availability of these nutrients is regulated by multiple nutrient-sensing pathways. These pathways act in a coordinated effort to regulate fetal growth and development, thereby ensuring adequate fetal nutrition, protection from maternal and cellular stress, and adequate oxygenation [2–7]. Two nutrient-sensing proteins involved in these processes are O-linked N-acetylglucosamine transferase (OGT) and mechanistic target of rapamycin (mTOR) kinase.

OGT is the sole enzyme that can add an O-GlcNac modification onto serine/threonine residues of target proteins to modulate their function, localization, and stability. As a nutri-

ent sensor protein, OGT regulates cellular growth and metabolism. OGA (O-GlcNAcase), on the other hand, removes the O-GlcNac modification from post-translationally modified proteins. In order to form O-GlcNAc, the substrate used by OGT, placental glucose is shuttled through the hexosamine biosynthetic pathway in which GFAT, or glutamine fructose-6-phosphate amidotransferase, is the rate-limiting enzyme. O-GlcNAcylation by OGT can be detected by anti-O-Linked N-Acetylglucosamine antibody (RL-2) and is used to examine the various effects of OGT on downstream products. OGT has been recently described as a placental biomarker for maternal stress, which likely affects fetal brain development [8] and metabolism in adulthood [9].

Mechanistic target of rapamycin, or mTOR, is a signal transduction protein kinase shown to alter the trajectory of fetal growth. mTOR exists in two complexes, mTORC 1 and mTORC 2. As a nutrient sensor, mTOR recognizes upstream triggers, such as nutrient availability, oxygen content, and hormonal signals (i.e., insulin), to control gene translation and protein synthesis. mTOR signaling is implicated in the regulation of placental function, including amino acid transport, in human and rodent models of FGR [10,11]. Mice with a placental mTOR reduction were found to exhibit fetal growth restriction and altered amino acid transport [11]. The role of mTOR in fetal growth is also demonstrated in fetal overgrowth, associated with increased signaling, as seen in obesity [12]. These findings and others support an increasingly diverse body of research regarding the role of placental nutrient sensors in regulating fetal growth, which allow for further studies and insight into the developmental origins of health and disease.

Maternal under- or over-nutrition during pregnancy has been widely thought to impact offspring metabolic health trajectory [13–17]. Prenatal exposure to nutrient-stress in the intrauterine environment may affect placental nutrient sensing and, subsequently, fetal growth. Both mTOR and OGT have been implicated to regulate human placental sensing and function. Therefore, we hypothesized that the gene and protein levels of mTOR and OGT are altered between FGR and control placentae. As such, the objective of this study is to evaluate the expression of two major nutrient sensor proteins, OGT, as well as its activity measure by RL2, and mTOR, along with and its downstream target pS6 S240 and pAKT S473, in the placentae of pregnancies complicated by FGR and appropriate for gestational age (AGA) placentae. We also aimed to assess whether the expression of these target proteins is impacted by fetal sex and mode of delivery in the AGA placentae.

## 2. Results

### 2.1. Birth Weight and Placental Weight Are Lower in Fetal Growth-Restricted Neonates Compared to Appropriate for Gestational Age Neonates

Maternal and fetal characteristics were analyzed for significant differences between AGA and FGR patients. There was no significant difference in maternal age of AGA and FGR neonates ($p = 0.6453$; Table 1). Although maternal BMI was also similar statistically between the AGA and FGR groups ($p = 0.1000$; Table 1), it is important to note that BMI lower than 25 is considered normal weight and 28 is over-weight. Samples from 14 AGA and six FGR pregnancies were obtained. Despite an attempt to collect an equal number for each sex, all six placentae of growth-restricted neonates were female. FGR occurred in three out of six mothers of Black or African American descent compared to one out of 14 mothers in the AGA group. Ten out of 14 mothers who gave birth to AGA neonates self-identified as White. Of the AGA pregnancies, nine were female and five were male offspring (Table 2). Mean gestational age at delivery was similar between AGA and FGR neonates ($p = 0.5225$; Table 2). Birth weight and placental weight were significantly lower in our FGR patients compared to our AGA controls ($p < 0.0001$ and $p = 0.0154$, respectively; Table 2). Placental volume showed a trending decrease in the FGR group but did not reach statistical significance ($p = 0.0688$; Table 2).

**Table 1.** Maternal characteristics, including mean maternal age, BMI, ethnicity, number of prior births, and mode of delivery.

| | | AGA (n = 14) | FGR (n = 6) | *p*-Value |
|---|---|---|---|---|
| **Maternal Age (years)** | | 30.29 ± 4.77 | 28.83 ± 8.25 | *p* = 0.6453 |
| **Maternal BMI** (Measure at pre-pregnancy or at new obstetric visit (NOB)) | | 24.43 ± 2.44 | 28.6 ± 7.08 | *p* = 0.1000 |
| **Maternal Ethnicity** | Black/African American | 1 | 3 | |
| | Asian | 2 | 0 | |
| | White | 10 | 2 | |
| | Hispanic (White) | 1 | 0 | |
| | "Other" | 0 | 1 | |
| **# of previous births** | 0 | 1 | 2 | |
| | 1 | 8 | 3 | |
| | 2 | 5 | 1 | |
| **Mode of delivery** | Cesarean | 10 | 5 | |
| | Vaginal | 4 | 1 | |

**Table 2.** Fetal characteristics, including gestational age, sex, birth weight, placental weight, and placental volume.

| | | AGA (n = 14) | FGR (n = 6) | *p*-Value |
|---|---|---|---|---|
| **Gestational Age (weeks)** | | 38.73 ± 1.13 | 38.36 ± 1.11 | *p* = 0.5225 |
| **Fetal Sex** | Male | 5 | 0 | |
| | Female | 9 | 6 | |
| **Birth weight (g)** | | 3357.5 ± 314.97 | 2364.33 ± 137.38 | *p* < 0.0001 |
| **Placental weight (g)** | | 561.39 ± 108.70 | 402.8 ± 76.79 | *p* = 0.0154 |
| **Placental volume (cm$^3$)** | | 1143.69 ± 482.58 | 586.93 ± 177.41 | *p* = 0.0688 |

*2.2. OGT and mTOR Levels in Female FGR Placentae Are Similar When Compared to Female AGA Placentae*

There was no significant difference in OGT or OGA gene expression between AGA and FGR placentae (Figure 1A,B). There was also no difference in GFAT, EMeg32, and FBPase between the two groups (Supplementary Figure S1A–C). Analysis of protein expression of OGT or OGA by Western blot revealed no significant difference between female AGA and FGR placentae (Figure 1C–E). Levels of O-GlcNacylation, measured by RL2, were also unchanged between female AGA and FGR placentae (Figure 1C,F). Expression and localization of mTOR and pS6 S240 were assessed via immunohistochemistry in female AGA and FGR placentae. Immunostaining of Cytokeratin 7 was included as a marker for trophoblast cells [18] (Supplementary Figure S2A). Immunostaining and analysis of mTOR protein expression revealed that mTOR is expressed throughout the placenta (Supplementary Figure S2B). However, there was no significant difference in intensity in the female AGA placentae compared to female FGR placentae (Supplementary Figure S2B,D). This was further supported by Western blot from whole protein lysates, which showed comparable mTOR protein expression between female AGA and FGR placentae (Figure 1C,G). IHC staining of pS6 S240, a downstream target of mTORC1 and a proxy for mTORC1 activity, showed localization of pS6 S240 largely in the syncytiotrophoblast layer. Expression of pS6 S240, however, did not differ between the placentae of female AGA and FGR fetuses

(Supplementary Figure S2C,D). This was again confirmed by pS6 S240 Western blot analysis (Figure 1C,H). Total S6 levels were also not different in the two groups (Figure 1H). mTOR is activated by phosphorylation of AKT via the PI3K/AKT pathway. Assessment of pAKT S473 and total AKT protein expression showed no significant differences between female AGA and FGR placentae (Figure 1C,I).

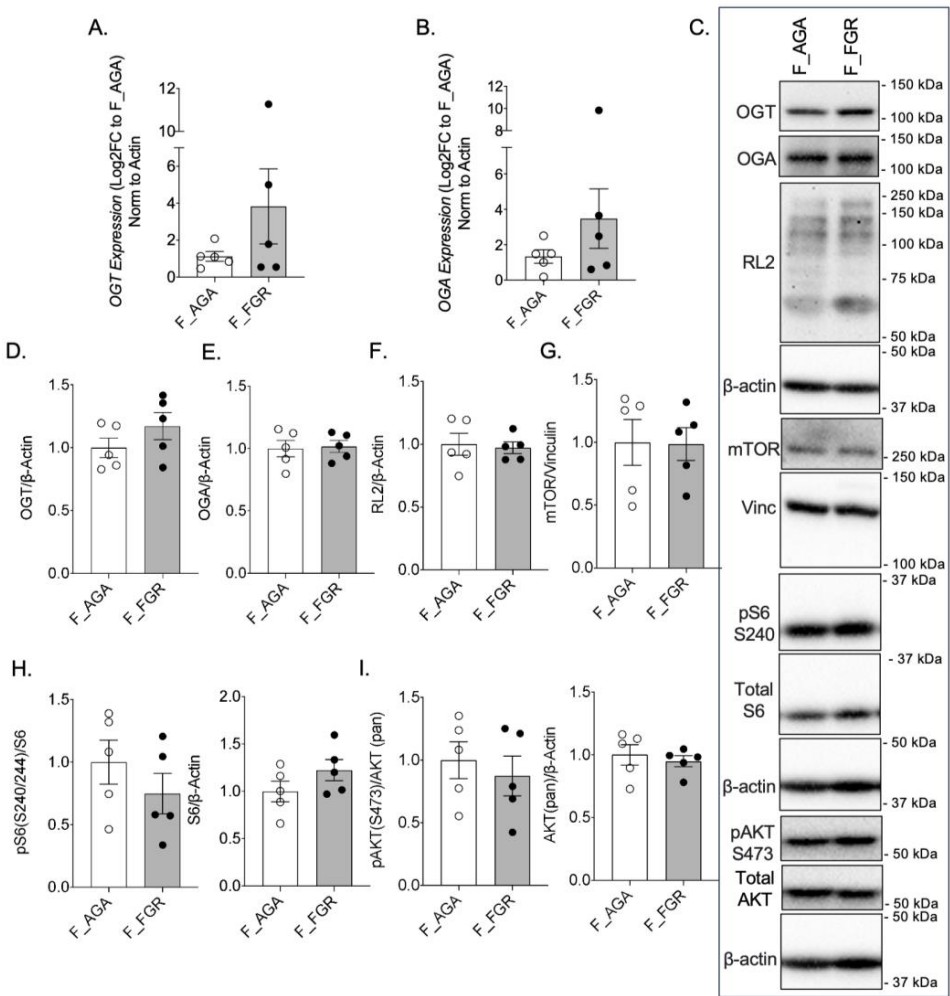

**Figure 1. OGT and mTOR levels are not altered in female FGR placentae.** (**A**) OGT and (**B**) OGA expression by qPCR in female AGA and FGR, cesarean delivery, placentae (n = 5/group). Analyses presented relative to β-actin, $log_2$-fold change to female AGA, cesarean delivery, placentae. (**C**) Representative immunoblots with densitometry measuring (**D**) OGT, (**E**) OGA, and (**F**) RL2, (**G**) mTOR, (**H**) pS6 S240, pan S6, and (**I**) pAKT S473, pan AKT relative to loading controls, vinculin or β-actin, from female AGA and female FGR, cesarean delivery, placentae (n = 5/group). Densitometry analyses presented relative to loading control proteins and normalized to female AGA, cesarean delivery, placentae. Values are reported as mean ± SEM. Statistical analyses were performed using unpaired, two-tailed Student's *t* test with $p < 0.05$.

### 2.3. In AGA Placentae, Expression of OGT and mTOR Are Unchanged Based on Fetal Sex

To examine whether the expression of nutrient sensor proteins is altered based on fetal sex, we compared male and female AGA placentae. There was no statistically significant difference in OGT or OGA mRNA levels between male AGA and female AGA placentae (Figure 2A,B). There was also no difference in GFAT, Emeg32, and FBPase between the two groups (Supplementary Figure S1D–F). Assessment of protein expression confirmed no differences in OGT and OGA in male AGA and female AGA placentae (Figure 2C–E). Levels of O-GlcNAcylation were also not significantly different between AGA placentae of male offspring when compared to AGA placentae of female offspring, further confirming our

findings (Figure 2C,F). Assessment of mTOR protein expression by IHC showed no difference in intensity between male and female AGA placentae (Supplementary Figure S3A,C). This was confirmed by Western blot of mTOR, which showed comparable protein levels between placentae of male and female AGA offspring (Figure 2C,G). There was also no significant difference in pS6 S240 protein between AGA male and AGA female placentae via IHC (Supplementary Figure S3B,C) or by Western blot from whole protein lysates of pS6 S240 and total S6 (Figure 2C,H). Finally, there was no difference in pAKT S473 and total AKT protein expression between male AGA and female AGA placentae (Figure 2C,I).

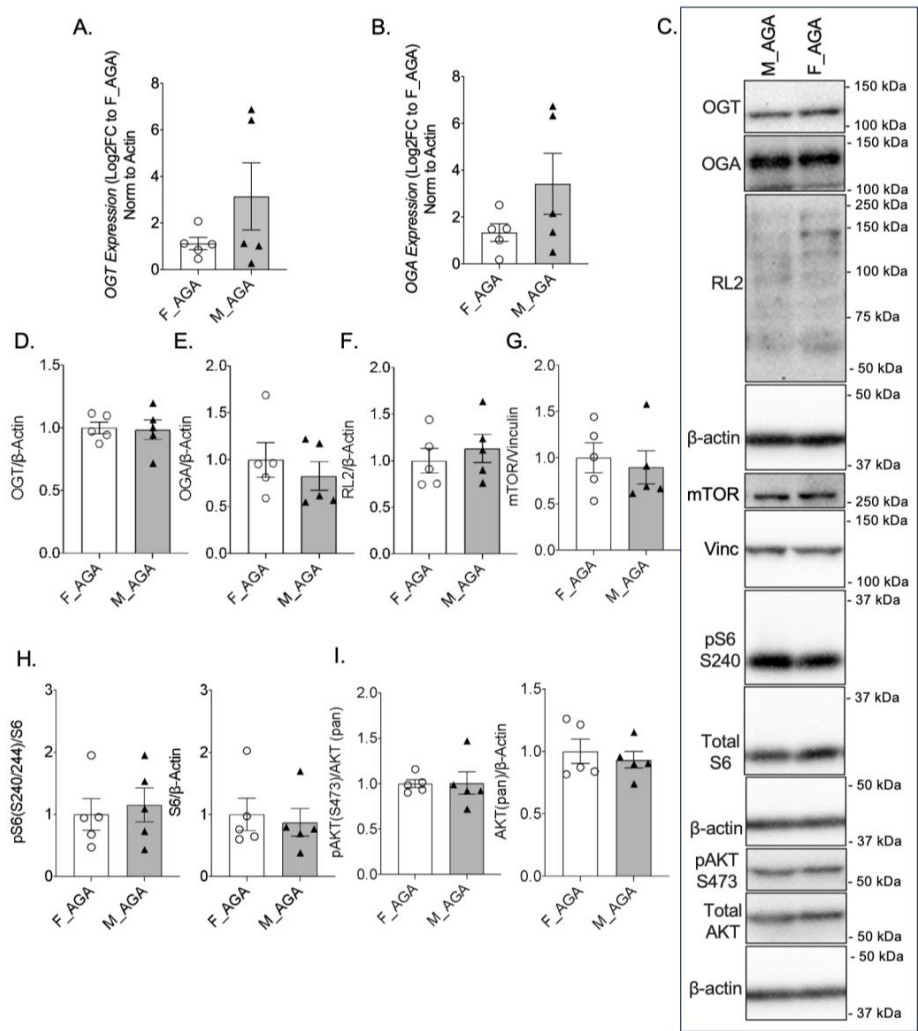

**Figure 2. OGT and mTOR levels are comparable in female and male AGA placentae.** qPCR gene expression of (**A**) OGT and (**B**) OGA in female AGA and male AGA, cesarean delivery, placentae (n = 5/group). Analyses presented relative to β-actin, $\log_2$-fold change to female AGA, cesarean delivery, placentae. (**C**) Representative western blots of OGT, OGA, RL2, mTOR, pS6 S240, S6, pAKT S473 and AKT, vinculin and β-actin from female AGA and male AGA, cesarean delivery, placentae. Bands presented for RL2 increased in contrast (across whole blot) post-quantification. Densitometry analyses presented for (**D**) OGT, (**E**) OGA, and (**F**) RL2, (**G**) mTOR, (**H**) pS6 S240, pan S6, and (**I**) pAKT S473, pan AKT relative to loading controls, vinculin or β-actin and normalized to female AGA, cesarean delivery, placentae (n = 5/group). Values are reported as mean ± SEM. Statistical analyses were performed using unpaired, two-tailed Student's *t* test with *p* < 0.05.

### 2.4. Mode of Delivery Does Not Alter the Expression of Proteins OGT and mTOR in Female AGA Placentae

To determine if the mode of delivery influences the expression and activity of OGT and mTOR, female AGA placentae from cesarean deliveries were compared to female AGA

placentae from vaginal deliveries. Between placentae of cesarean and vaginal deliveries, there was no statistical difference in OGT or OGA gene expression (Figure 3A,B). No statistically significant difference was observed in GFAT, EMeg32, and FBPase mRNA levels (Supplementary Figure S1G–I). Protein expression of OGT and OGA were also comparable between AGA placentae of cesarean and vaginal deliveries (Figure 3C–E). Assessment of mTOR by Western blot showed similar levels of mTOR expression in AGA female placentae from cesarean deliveries compared to AGA female placentae from vaginal deliveries (Figure 3C,F). Expression of pS6 S240 and total S6 also did not show a significant difference based on mode of delivery in the female AGA placentae (Figure 3C,G). Finally, pAKT S473 and total AKT protein expression in female AGA placentae after cesarean delivery when compared to vaginal delivery were similar (Figure 3C,H).

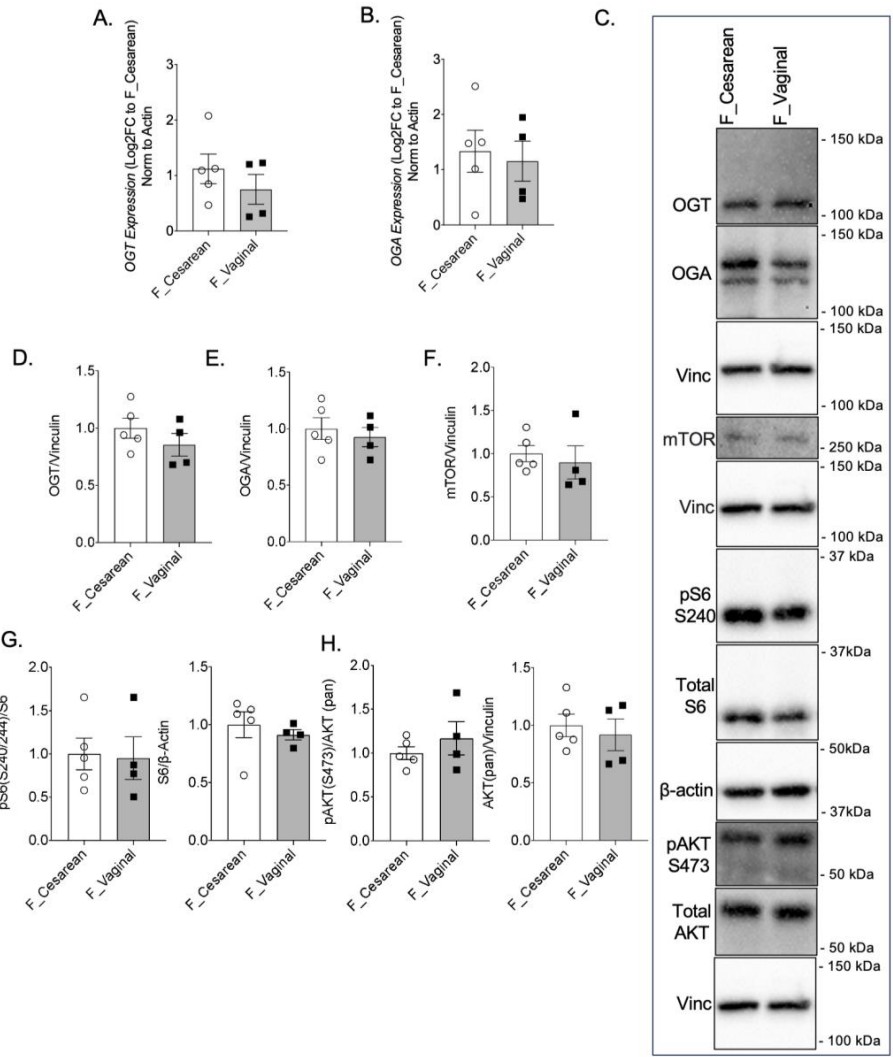

**Figure 3. Mode of delivery does not alter OGT and mTOR levels in AGA placentae.** (**A**) OGT and (**B**) OGA gene expression by qPCR in female AGA, cesarean delivery, placentae (n = 5) and female AGA, vaginal delivery, placentae (n = 4). Analyses presented relative to β-actin, $\log_2$-fold change to female AGA, cesarean delivery, placentae. (**C**) Representative immunoblots with densitometry measuring (**D**) OGT, (**E**) OGA, (**F**) mTOR, (**G**) pS6 S240, pan S6 and (**H**) pAKT S473, pan AKT relative to Vinculin, from female AGA, cesarean (n = 5) and vaginal delivery (n = 4), placentae. Densitometry analyses presented relative to vinculin or actin and normalized to female AGA, cesarean delivery, placentae. Values are reported as mean ± SEM. Statistical analyses were performed using unpaired, two-tailed Student's *t* test with $p < 0.05$.

## 3. Discussion

Fetal growth restriction is a pregnancy complication in which the fetus fails to achieve its expected growth potential [19]. While the etiology of FGR is complex, placental insufficiency and altered nutrient and oxygen delivery are major contributors to the development of FGR. OGT and mTOR are two nutrient-sensing proteins that regulate fetal growth and development and mediate nutrient flux to the fetus in response to nutrient availability [20]. One of the aims of this study was to assess differences in expression of OGT and mTOR in female AGA or FGR placentae and to examine whether the sex of the fetus played a role in the expression of these nutrient-sensor proteins in female and male AGA placentae. Finally, we investigated whether the mode of delivery influenced the expression of these nutrient-sensor proteins in female AGA placentae delivered by cesarean delivery or vaginal delivery.

Based on a previous study in a maternal corticosterone-exposed mouse model, we initially hypothesized that OGT, as a likely placental biomarker of maternal stress and nutrient status, would show decreased expression in the placentae of women with FGR when compared to placentae of women with AGA neonates [21]. In our current sample pool, we did not observe any differences in OGT, OGA, or RL2 levels in the female FGR placentae when compared to the female AGA placentae. Another study by Howerton et al. demonstrated that OGT levels are dependent on X chromosome dosage in their human placenta samples, which may indicate that OGT is differentially expressed in males [22]. When comparing female and male AGA placentae in our sample group, we found no significant differences in the expression of these proteins. Finally, we found that the mode of delivery did not alter the expression of OGT, OGA, and RL2 in our female AGA samples.

As a nutrient sensor, mechanistic target of rapamycin (mTOR) has been implicated to be involved in the trajectory of fetal growth [23], and previous studies have demonstrated that mTOR activity is reduced [19,20,24–27] or increased [28] in FGR placentae. While our placental staining confirmed the presence of mTOR and pS6 S240 in the human trophoblast cells [25,27,29], we observed no differences in mTOR protein expression, assessed both by IHC and western blotting, between our female AGA and FGR placentae. We also found that fetal sex and mode of delivery did not impact mTOR expression in our AGA samples. Similar findings were observed for the downstream target of mTORC1, pS6 S240, as well as the upstream regulator, pAKT S473. Our data are supported by Fahlbusch et al., where they reported similar findings to our current study [29]. They found that total and activated levels, as well as activated-to-total ratios of placental mTOR, p70S6K1, and AKT were not changed between their FGR and AGA samples. As reviewed by Dong et al. [30], previous studies have reported increased, decreased, or no change in the levels of mTOR in FGR placentae, which may partially be explained by differential methods of analyses in these studies. In a study by Roos et al., they found that placental expression of mTOR was increased in their intrauterine growth restriction (IUGR) samples, and the activity of mTOR, measured by protein expression of phosphorylated S6K1 Thr389, was reduced [25]. Another study by Tsai et al., examined the differential expression of activated mTOR in control and FGR placentae and observed reduced levels of activated mTOR in the FGR placentae [27]. They also examined p-p70 S6K and observed elevated levels of pS6 expression in FGR placentae when compared to controls. The difference between all these studies could possibly be explained by several factors, including variability in patient inclusion, sample procurement, and method of analysis (i.e., western blotting vs. IHC). The FGR samples collected in the study by Tsai et al., for example, were confirmed by ultrasound showing placental insufficiency with uterine Doppler, absent end diastolic flow, and an estimated fetal weight below the 10th percentile [27]. In our current study, while many of our FGR group were <3rd percentile in birth weight, none of the cohort exhibited umbilical artery Doppler changes that we normally associate with this condition. The current study is also limited by the fact that it can often be difficult to distinguish between SGA (small for gestational age infants) who are constitutionally small versus those who have true growth restriction. There may also be differences in gestational age and sample

handing during the collection, which may contribute to the differences in our findings. We also acknowledge that we are limited by our small sample size; thus, we speculate that it may also partially explain why we did not observe any differences in our samples.

Prenatal stress may be an important factor to consider, since prenatal exposure to stress in the intrauterine environment affects placental metabolism and subsequently fetal growth. Epidemiological studies have attributed an increased incidence of FGR with maternal stress caused by exposure to civil unrest and socioeconomic and ethnic inequalities [31,32]. mTOR and OGT nutrient-sensing proteins have also been shown to respond to various hypoxic and metabolic derangements related to fetal stress, serving as a protective compensatory mechanism under normal physiological conditions [5,8,33]. Furthermore, studies have demonstrated that levels of OGT and O-GlcNAcylation are not only reduced in placentae of male fetuses but are further reduced when exposed to prenatal stress [8,22,34–37]. The mode of delivery may also be important to consider when examining nutrient sensors and stress signaling in placentae samples. A previous study demonstrated that, in response to labor, total placental mTOR and phosphorylated mTOR were significantly reduced [38]. 11β-hydroxysteroid dehydrogenase-2 (11β-HSD2) is a well-established enzyme involved in the placental stress pathway and serves to protect the developing fetus from maternal stress [39]. Studies have shown a decrease in 11β-HSD2 activity in human and rodent FGR models [21,40–43]. Our female AGA samples did not differ in OGT and mTOR levels or activity based on mode of delivery. Furthermore, assessment of 11β-HSD2 in our samples did not reveal any statistically significant differences, except for mRNA expression, when comparing mode of delivery in our female AGA placentae ($p$ = 0.0401) (Supplementary Figure S4). There were also no changes in protein levels of 11β-HSD2.

## 4. Materials and Methods

This was a translational descriptive study that occurred in a large academic tertiary care hospital following IRB approval at the University of Minnesota. A total of 20 participants were recruited for the study. Participants were recruited if the fetus was diagnosed with FGR, defined as an estimated fetal weight and abdominal circumference less than the 10th percentile [1,44]. From March 2018 until March 2022, a total of n = 6 placentae were collected from FGR fetuses, with the aim to match placentae from AGA fetuses (n = 14) in a 2:1 ratio. The initial recruitment goal of obtaining six FGR samples was based upon the ability to detect a 30% difference in OGT protein expression on Western Blot and power analysis of 0.8, based on earlier work by Alejandro et al. in mice [45]. Neonatal birthweights above the 10th percentile were excluded. All healthy women without underlying maternal disease were included. Demographic information, fetal sex, and mode of delivery were recorded. Women with pregnancies complicated by congenital anomalies, multifetal gestation, genetic abnormalities, congenital infection, and known placental or umbilical cord anomalies were excluded, as were women with diabetes, hypertension, obesity, or COVID-19.

### 4.1. Sample Procurement

Placental samples were collected in accordance with recommendations put forth by Burton et al. [46]. Briefly, samples of approximately 5 g were collected from the midsection of the placenta, and neither the cord nor the periphery. Biopsies were taken in quadruplicate—three randomly from the body of the placenta and one sample taken near the umbilical cord insertion [46]. Four 1 cm$^3$ sections were taken from each of four quadrants of the placenta, coming from the middle of middle cotyledons with neither the maternal nor fetal membrane present. Placental samples were washed in PBS (phosphate buffered saline) to remove maternal contaminate and snap frozen within 30 min of placental separation at delivery. Placental volume was measured along three planes for each placenta post-delivery, per Burton et al. [46]. All tissue samples were taken in accordance with oversight of Bionet, the tissue procurement arm of the Clinical and Translational Science Institute at the University of Minnesota.

### 4.2. RT qPCR

RT qPCR was performed for mRNA levels of the genes involved in the hexosamine biosynthetic pathway (HBP) and glycolysis: OGT, OGA, GFAT, glucosamine-6-phosphate (GlcN6P) acetyltransferase (EMeg32), fructose 1,6-bisphosphatase (FBPase), 11β-HSD2, and β-actin. Primer sequences are presented in Supplementary Table S1. RNA for gene expression analysis was isolated using an Rneasy kit (Qiagen; Hilden, Germany). RNA concentration was assessed using TECAN InfinitePro Plate Reader software (Magellan 7.2) (Männedorf, Switzerland). cDNA synthesis was performed using random hexamers and was reverse transcribed using SuperScript II (Life Technologies; Carlsbad, CA, USA) according to the manufacturer's protocol. Real-time PCR was performed on an ABI 7000 sequence detection system using SYBR Green (Applied Biosystems; Waltham, MA, USA).

### 4.3. Western Blotting

Placental samples were probed for OGT, mTOR, and key proteins involved in the nutrient-sensing pathways. Briefly, protein lysates were prepared by thawing samples and adding lysis buffer with protease and phosphatase inhibitor cocktail (ABCAM; Cambridge, United Kingdom). To prevent the removal of O-GlcNAc from proteins by OGA, 1 μL of 10 μM Thiamet-G (TMG), an OGA inhibitor (Carbosynth; Staad, Switzerland), was added to the lysis buffer. Protein concentrations were determined using the BCA protein assay with BSA as the standards. Protein lysates (50 μg) were separated by SDS-Page and run on 7% and 10% gels. Electrophoresis was performed at a constant Amp of 0.04/gel for approximately 90 min and proteins were transferred to PVDF membranes overnight at a constant Amp of 0.07 at 4 °C. Membranes were blocked in 5% milk in Tris-buffered saline in room temperature for one hour, followed by overnight primary antibody incubation at 4 °C. Whole membranes were cut to allow for multiple protein detections. Sources of primary antibodies and corresponding dilution factors are presented in Supplementary Table S2. Membranes were incubated with primary antibodies against mTOR (1:1000, 2983S, Cell Signaling; Danvers, MA, USA) and OGT (1:1000, 24083S, Cell Signaling). To assess OGT and mTOR activity, membranes were incubated with mouse monoclonal RL2 to O-Linked N-Acetylglucosamine (1:1000, ab2739, ABCAM) for presence of O-GlcNAcylation, as well as antibodies against OGA (1:500, SAB4200267 and SAB4200311, Sigma; St. Louis, MO, USA), pS6 S240 (1:500, 5364S, Cell Signaling) and total S6 (1:1000 and 1:500, sc-74459, Santa Cruz), pAKT S473 (1:1000 and 1:500, 4060S, Cell Signaling), and total AKT (1:500, 2920S, Cell Signaling). Vinculin (1:1000, 13901S, Cell Signaling; 1:500, 66305-1-Ig, Proteintech; Rosemont, IL, USA) and β-actin (1:1000, 3700S, Cell Signaling) were used as loading controls. We note that there was no statistically significant difference in Vinculin or β-actin loading controls when comparing any sample group. Secondary antibodies conjugated to horseradish peroxidase were purchased from Jackson Immunoresearch. Densitometry analysis was performed using ImageJ software 1.53a as previously described [47] and all values were normalized to the mean density of the female AGA placentae.

### 4.4. Immunohistochemistry and Analysis

Formalin-fixed, paraffin-embedded placentae were sectioned and immunostained using a Rabbit specific HRP/DAB (ABC) Detection IHC Kit (ab64261), per the manufacturer's instructions. Sections were incubated with primary antibodies overnight, followed by HRP-labeled-streptavidin and a biotinylated anti-rabbit secondary antibody. Primary antibodies with corresponding dilution factors can be found in Supplementary Table S2. The following primary antibodies with its corresponding dilution factors was used: mTOR (1:500 and 1:200, 2983S, Cell Signaling) and pS6 S240 (1:500 and 1:200, 5364S, Cell Signaling). Sections were counterstained with hematoxylin. Cytokeratin 7 (1:500, ab181598, ABCAM) was stained to characterize trophoblast cells (Supplementary Figure S2A). Analysis of protein expression levels through chromogen immunostaining intensity was performed using ImageJ software as previously described [48]. Three images per placentae sample

were randomly captured at 20× magnification. The intensity was calculated and divided by the number of nuclei per image, and the average of the three images was taken.

### 4.5. Statistical Analysis

The following comparisons were made in this study: (1) female AGA cesarean placentae vs. female FGR cesarean placentae, (2) female AGA cesarean placentae vs. male AGA cesarean placentae, and (3) female AGA cesarean placentae vs. female AGA vaginal placentae. Data are presented as mean $\pm$ SEM. Data were analyzed by a 2-tailed Student's *t*-test using GraphPad Prism version 8 (GraphPad Software), where appropriate. Results were considered statistically significant when $p < 0.05$.

## 5. Conclusions

In this present study, we did not observe significant differences in the expression of nutrient sensor proteins, OGT and mTOR, and their signaling activities in any of our sample groups. Future studies may consider assessing expression levels of OGT, mTOR, and key proteins involved in their signaling pathways in specific cell types (i.e., syncytiotrophoblast and cytotrophoblast) for potential cell-specific differences and in various conditions (i.e., fasting vs. fed states). We acknowledge that our study is limited by the fact that all our FGR/AGA placentae were female. Because much of what we know about sexual dimorphism and fetal growth is biased towards male offspring [22,34–37], this bias may have also affected our findings. Nevertheless, the current data offer unbiased results and present a greater picture of the complexity of placental gene expressions of OGT and mTOR.

**Supplementary Materials:** The following supporting information can be downloaded at: https://www.mdpi.com/article/10.3390/stresses4020019/s1, Table S1: List of primer sequences for qPCR. Table S2: List of antibodies used for western blot and immunohistochemistry analyses. Supplementary Figure S1. qPCR gene expression for GFAT, EMeg32, and FBPase in female AGA and FGR placentae, female and male AGA placentae, and cesarean and vaginal delivery. Supplementary Figure S2. Representative IHC staining of female AGA and FGR, cesarean delivery, placentae. Supplementary Figure S3. Representative IHC staining of female AGA and male AGA, cesarean delivery, placentae. Supplementary Figure S4. Assessment of placental 11β-HSD2 expression by qPCR in female AGA and FGR placentae, female and male AGA placentae, and cesarean and vaginal delivery placentae.

**Author Contributions:** Conceptualization, E.M. and E.U.A.; methodology, E.M., G.C., S.J. and B.C.; validation, and E.M., G.C., S.J. and B.C.; formal analysis, E.M., G.C., S.J. and B.C.; resources, E.M., S.A.W. and E.U.A.; data curation, E.M., G.C., S.J. and B.C.; writing—original draft preparation, E.M., G.C. and E.U.A.; writing—review and editing, E.M., G.C., E.U.A., S.A.W., S.J. and B.C.; supervision, E.U.A.; funding acquisition, E.U.A. and S.A.W. All authors have read and agreed to the published version of the manuscript.

**Funding:** This work was supported by the National Institutes of Health Grant NIDDK R56DK131447, R01DK136237, R01DK115720 to EUA, F31DK131860 to SJ and AHA-UPRIME (24IAUST1191276) Award for BC. This research was supported by the National Institutes of Health's National Center for Advancing Translational Sciences, grant UL1TR002494. The content is solely the responsibility of the authors and does not necessarily represent the official views of the National Institutes of Health's National Center for Advancing Translational Sciences. SAW is funded by the NICHD Reproductive Scientist Development Program, (K12 HD000849); EM was supported by the Department of Obstetrics, Gynecology and Women's Health. EUA and SAW were supported by Minnesota Institute of Diabetes, Obesity and Metabolism.

**Institutional Review Board Statement:** Institutional Review Board at the University of Minnesota approved this study (00033282 and 00002076).

**Informed Consent Statement:** Informed consent was obtained from all subjects involved in the study.

**Data Availability Statement:** Data available at request.

**Acknowledgments:** We would also like to extend our gratitude to all the hospital staff in the Department of Obstetrics, Gynecology and Women's Health and the patients for their participation

and contribution to this study. We thank Bethany Hart and CTSI at UMN for assistance in the IRB application.

**Conflicts of Interest:** The authors declare no conflicts of interest.

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
