# Peer review of "Gene and Protein Expression of Placental Nutrient-Stress Sensor Proteins in Fetal Growth Restriction"

_stresses, doi:10.3390/stresses4020019_

Round 1

Reviewer 1 Report

Comments and Suggestions for Authors

The article “Gene and protein expression of placental nutrient-stress sensor proteins in fetal growth restriction” by Morgan et al. is a relatively simple study that sheds light on the expression two stress sensing proteins OGT and mTOR kinase in fetal growth restricted placentae. The manuscript is well written, but the outcome is not what the authors expected and argued for as the premise to the manuscript. Nevertheless, these results are important and show that what’s generally presumed might not always be the case. As a big negative for me, the main figures are unnecessarily stretched out to make the manuscript appear larger than it actually is. The authors, at some point, also argue that the sample size was too small for this study. It is understandable that its not easy to procure these samples and the comparisons made with what the authors had at hand seem to be sufficient for the findings. This is of course keeping in mind the nature and extent of the data that is presented.

The article needs a major overhaul in the way the data is presented in its figures, and as such, it needs a major revision before it can be considered for publication.

Major comments –

1.     The Abstract should be short and free flowing. In its current iteration, it is too long and contains unnecessary sections like materials and methods. Please reduce the abstract to have an introduction followed by your findings and perspectives (if necessary) that are free flowing in one continuous paragraph.

2.     I do not understand why the authors chose to do 6 separate figures. Figures pairs 1 & 2, 3 & 4, and 5 & 6 contain the exact same information when paired respectively, the only difference being the proteins considered - OGT/OGA or mTOR. As such, the figures should be downsized, fusing 1 & 2, 3 & 4, and 5 & 6 (i.e. 1 & 2 combined = Fig 1, 3 & 4 combined = Fig 2). Each title of the new figure should read something in the lines of “OGT, O-GlcNAc and mTOR levels are not altered in……..”

Minor comments –

1.     There are some typographical errors in the manuscript and figure legends that need to be corrected. Please read the text thoroughly and you will find the errors.

Comments on the Quality of English Language

1.     There are some typographical errors in the manuscript and figure legends that need to be corrected. Please read the text thoroughly and you will find the errors.

Author Response

We would like to thank you for taking the time to review our manuscript and for your positive comments suggestions to improve it.

Major Comments

  1. The Abstract should be short and free flowing. In its current iteration, it is too long and contains unnecessary sections like materials and methods. Please reduce the abstract to have an introduction followed by your findings and perspectives (if necessary) that are free flowing in one continuous paragraph.

Thank you for this suggestion, and we agree, the abstract has been shortened.

  1. I do not understand why the authors chose to do 6 separate figures. Figures pairs 1 & 2, 3 & 4, and 5 & 6 contain the exact same information when paired respectively, the only difference being the proteins considered - OGT/OGA or mTOR. As such, the figures should be downsized, fusing 1 & 2, 3 & 4, and 5 & 6 (i.e. 1 & 2 combined = Fig 1, 3 & 4 combined = Fig 2). Each title of the new figure should read something in the lines of “OGT, O-GlcNAc and mTOR levels are not altered in……..”

As suggested, we’ve consolidated figures, we now have only 3 figures, and we also shortened the result section and combine when necessary for flow and clarity.

Minor Comments

  1. There are some typographical errors in the manuscript and figure legends that need to be corrected. Please read the text thoroughly and you will find the errors.

We appreciate this comment, we have now review the manuscript and figure for grammar and clarity.

Reviewer 2 Report

Comments and Suggestions for Authors

The study “Gene and protein expression of placental nutrient-stress sensor proteins in fetal growth restriction” by Morgan et al. investigated the link between the effect of restricted fetal growth and the gene expression and protein levels of mTOR and OGT.

Special attention was given to determine mRNA and protein expression of OGA, RL2, mTOR, pS6 S240, and pAKT S473 and to the localization of these proteins in the placenta. Furthermore, the mode of delivery and the fetal sex were analyzed.

There is evidence that mTOR plays a key role in the utero-placental blood flow, placental function, fetal growth and fetal programming. OGT play an important role in maternal homeostasis, energy supply and modifies transplacental signals for the fetal development.The reviewer appreciates this smart logical approach and the used method which are very appropriate to study this important phenomenon.

Major concerns:

The authors did not find any significant differences in the expression of OGT, mTOR, and their signaling activities. Nevertheless, the reviewer agrees with their statement that the results presents a greater picture of the complexity of placental gene expressions of OGT and mTOR. However, they should find a better balance between their findings and their extended reporting molecular content.

The authors analyzed the molecular results also regarding the ethnic differences; however without any scientific approach and profound results. Therefore, the conceptual approach and their statements are overstated and speculative.

L 185 and table 2: The authors mentioned that the BMI is similar in both groups. From a statistical point, based on a 2-tailed Student’s test. There are no objections – however a physiological view supports also more possible metabolic interactions. Furthermore, the authors should reconsider their statistical approach.

Minor aspects:

Table 1 should be reworked.

No units of measurement for maternal age (Years?) and gestational age (weeks ?)

Author Response

We thank you for your time and positive feedback. We appreciate your comments about the logical approach and the method we used in the current study.

Major Comments:

The authors did not find any significant differences in the expression of OGT, mTOR, and their signaling activities. Nevertheless, the reviewer agrees with their statement that the results presents a greater picture of the complexity of placental gene expressions of OGT and mTOR. However, they should find a better balance between their findings and their extended reporting molecular content.

We have now reduced the length of the manuscript, and shorten our statements for clarity.

We agree, and we have now reduce

L 185 and table 2: The authors mentioned that the BMI is similar in both groups. From a statistical point, based on a 2-tailed Student’s test. There are no objections – however a physiological view supports also more possible metabolic interactions. Furthermore, the authors should reconsider their statistical approach

We understand the concern here regarding the statistical point on BMI. Therefore, we have now revised the manuscript to state, “Although maternal BMI was also similar statistically between the AGA and FGR groups (p=0.1000; Table 1), it is important to note that BMI lower than 25 is considered normal weight and 28 is over-weight”. We reconsidered our statistical approach on BMI. However, since we are comparing the means of BMI between two groups only, we employed the independent samples t-test and not the paired samples t-test.

Minor Comments:

Table 1 should be reworked.

No units of measurement for maternal age (Years?) and gestational age (weeks ?)

Thank you for pointing this out. We’ve clarified texts and units were updated. 

Thank you, and we appreciate your comment about the logical approach and the method we used.

Major Comments:

The authors did not find any significant differences in the expression of OGT, mTOR, and their signaling activities. Nevertheless, the reviewer agrees with their statement that the results presents a greater picture of the complexity of placental gene expressions of OGT and mTOR. However, they should find a better balance between their findings and their extended reporting molecular content.

*We have now reduced the length of the manuscript, and shorten our statement for clarity. 

The authors analyzed the molecular results also regarding the ethnic differences; however without any scientific approach and profound results. Therefore, the conceptual approach and their statements are overstated and speculative.

We agree, and we have now taken this out in the discussion. When appropriate, we also stated if we are speculating.

L 185 and table 2: The authors mentioned that the BMI is similar in both groups. From a statistical point, based on a 2-tailed Student’s test. There are no objections – however a physiological view supports also more possible metabolic interactions. Furthermore, the authors should reconsider their statistical approach

*We understand the concern here regarding the statistical point on BMI. Therefore, we have now revised the manuscript to state, “Although maternal BMI was also similar statistically between the AGA and FGR groups (p=0.1000; Table 1), it is important to note that BMI lower than 25 is considered normal weight and 28 is over-weight”. We reconsidered our statistical approach on BMI. However, since we are comparing the means of BMI between two groups only, we employed the independent samples t-test and not the paired samples t-test.

Minor Comments:

Table 1 should be reworked.

No units of measurement for maternal age (Years?) and gestational age (weeks ?)

Thank you for pointing this out. We’ve clarified texts and units were updated. 

Round 2

Reviewer 1 Report

Comments and Suggestions for Authors

I am satisfied with the edits that the authors have made. I am happy to recommend this manuscript for publication at this stage.